# Does Environmental Regulation Improve the Green Total Factor Productivity of Chinese Cities? A Threshold Effect Analysis Based on the Economic Development Level

**DOI:** 10.3390/ijerph18094828

**Published:** 2021-04-30

**Authors:** Xinfei Li, Chang Xu, Baodong Cheng, Jingyang Duan, Yueming Li

**Affiliations:** School of Economics and Management, Beijing Forestry University, Beijing 100083, China; lxfbjfu@163.com (X.L.); xuchang@bjfu.edu.cn (C.X.); duanjingyang0526@163.com (J.D.); lym_aria98@bjfu.edu.cn (Y.L.)

**Keywords:** environmental regulation, green total factor productivity, China, threshold model

## Abstract

Improvement of green total factor productivity (GTFP) through environmental regulation is of great practical significance in promoting the high-quality development of urban economies. Based on panel data for 163 cities in China from 2003 to 2016, five indicators were selected to quantify the effects of environmental regulation: the SO_2_ removal rate, smoke and dust removal rate, solid waste utilization rate, domestic sewage treatment rate, and waste harmless treatment rate. This study evaluated the impact of environmental regulation on urban GTFP, and analyzed the threshold effect of urban economic development levels. The research results showed that the impact of environmental regulations on GTFP changed as the level of urban economic development increases. When the economic development level was low, environmental regulation had a significant positive effect on GTFP, especially the SO_2_ removal rate. When the economy developed to reach a medium level, the impact of environmental regulation on GTFP was negative. When the economic development level was high, the SO_2_ removal rate still had a significant positive impact on GTFP. The solid waste utilization rate had a significant negative impact on GTFP. It was concluded that the government should consider the local economic development level when formulating environmental regulation policies.

## 1. Background and Introduction

Although great achievements have been made in China’s urban economic development, problems of resource shortages and environmental deterioration have emerged [1]. In recent years, haze and water pollution issues have occurred in most Chinese cities. The increasingly severe pressure on resources and the environment has led to concerns about the sustainability of China’s economic development. The traditional indicator of regional economic performance is GDP (Gross Domestic Product) [2]. Gross Domestic Product (GDP) refers to the value of all the final products and services produced in the economy of a country or region in a certain period of time (a quarter or a year), which explains why some cities blindly pursue a high-yield development mode [3]. In response to this situation, the Chinese government has begun to promote green development [4], with a change in emphasis from pursuing GDP growth to pursuing green total factor productivity (GTFP) [5]. Green total productivity refers to the total factor productivity estimated by taking pollutant emissions as unpaid inputs and introducing them into the production function together with capital, labor, and energy inputs [6,7,8,9,10]. To improve GTFP, the Chinese government has formulated and improved a series of environmental policies to promote high-quality economic development [11], such as “ambient air quality standard” and “air pollution prevention and control law”.

There have been relatively few studies of the relationship between environmental regulation and GTFP at the urban level. Moreover, the existing research on the relationship between environmental regulation and GTFP has not reached a consistent conclusion. In previous studies, three different hypotheses have been proposed regarding the impact of environmental regulation on GTFP. (1) Environmental regulation hinders the improvement of GTFP. Strict environmental regulation would increase the pollution control costs for local enterprises and inhibit their research and development (R&D) and innovation activities. In turn, the GTFP declines [12,13,14,15,16,17,18]. In addition, some researchers have investigated the “pollution paradise hypothesis”. The “pollution paradise hypothesis” mainly means that enterprises in pollution-intensive industries tend to be established in countries or regions with relatively low environmental standards [19]. They found that after the improvement of local environmental regulations, pollution-intensive enterprises would transfer to other areas to avoid the high production costs, and the transfer would reduce the local GTFP to a certain extent [20]. (2) Environmental regulation promotes the improvement of GTFP. It has also been claimed that environmental regulation can promote regional technological innovation, offset the cost of environmental governance, and then improve GTFP [21,22,23]. Naso et al. [24] reported that the enhancement of environmental regulation intensity would “force” enterprises to increase research and development (R&D) in terms of energy consumption and emission reduction. In this way, the competitiveness of enterprises could be improved to promote the urban GTFP [25,26,27]. (3) There is a non-linear relationship between environmental regulations and GTFP [28,29,30]. In addition, some researchers believe that there are regional differences in the impact of environmental regulation on GTFP [31]. Ren [32] found that with the increase of environmental regulation intensity in China, the GTFP in the eastern region displayed an upward trend, while in the central and western regions, there was a downward trend. There are three reasons why researchers have not reached a consensus. The first issue: different researchers have used different samples in their research. Some researchers have conducted studies at the national level, while others have considered the provincial level, which is an approach that is prone to errors [33,34,35]. The second issue: research on environmental regulation mostly started from a certain regulatory policy [36,37], and few studies analyzed from the perspective of the effect of environmental regulation, such as the effect of pollutant emission reduction. The third issue: The effect of differences in the economic development level on the effect of environmental regulation has been ignored. Differences in economic development level often determine the policy objectives of local governments, which influence the effectiveness of environmental regulation [38].

To make up for the deficiencies of existing research, this article makes the following improvements. First of all, in response to the problem of regional errors caused by the selection of provincial samples for empirical analysis, this article chooses to use prefecture-level city data for analysis. The prefecture-level city data are more representative, describing in detail the strength of China’s environmental regulations and GTFP. Secondly, past studies have been difficult to reflect the effects of environmental regulations. Therefore, this paper selects industrial SO_2_ removal rate, soot removal rate, comprehensive utilization rate of industrial solid waste, domestic sewage treatment rate, and domestic waste. We used five indicators of the rate of harm treatment. Further, we used the entropy method to calculate the comprehensive index of the environmental regulation intensity of the synthetic urban agglomeration, in order to make a comprehensive explanation of the intensity of environmental regulation in Chinese cities. Finally, in view of the lack of consideration of the existing research on whether the urban economy affects the effect of environmental regulation, this paper constructs a threshold model for environmental regulation and GTFP, and selects per capita GDP as the threshold variable.

This paper has research contributions in the following three aspects: the first contribution: based on city-level data, this paper uses the panel threshold model to more delicately measure the impact of environmental regulation intensity on GTFP. This study expands the empirical evidence to study the impact of environmental regulations on GTFP from the city level. The second contribution: based on the effect of urban pollutant emission reduction, this paper analyzes environmental regulations and the effects of different pollutants in environmental regulations on GTFP. This helps to understand the impact of environmental regulations on GTFP more comprehensively. The third contribution: this article examines the impact of environmental regulations on GTFP under different economic levels. This can help city managers formulate more targeted strategies to help improve the city’s GTFP and take the path of sustainable development.

The remaining chapters of this paper are arranged as follows: Section 2 is the research design, which includes the model design and variable selection; Section 3 presents the analysis of the empirical results; Section 4 is the discussion and the conclusion to provide suggestions for further study.

## 2. Research Design

### 2.1. Construction of the Measurement Model

To explore the non-linear relationship between environmental regulation and GTFP due to differences in urban economic development levels, the panel threshold regression method developed by Hansen [39] is used to estimate the parameters. The model is expressed as follows:(1)$$Y_{i,t}=c+\beta_{1}X_{i,t}I\left(q_{i,t}\leq \gamma \right)+\beta_{2}X_{i,t}I\left(q_{i,t}>\gamma \right)+{\theta X}_{i,t}+u_{i}+e_{i,t}$$
where i represents the city; t represents the year; $X_{i,t}$ is the explanatory variable; $Y_{i,t}$ is the explained variable; $q_{i,t}$ is the threshold variable; γ is the unknown threshold value; $e_{i,t}$ is the random disturbance item, $e_{i,t}\sim \mathrm{idd}\;N\left(0,\sigma^{2}\right)$; $u_{i}$ is the individual effect; I(·) is the indicator function.

To analyze the non-linear impact of environmental regulations on GTFP, per capita GDP is used as the threshold variable. A single-threshold panel model is constructed between GDP and GTFP. The following panel threshold model is constructed:(2)$${\mathrm{GTFP}}_{i,t}=c+\beta_{1}{\mathrm{REGU}}_{i,t}I\left(q_{i,t}\leq \gamma_{1}\right)+\beta_{2}{\mathrm{REGU}}_{i,t}I\left(\gamma_{1}<q_{i,t}<\gamma_{2}\right)+\cdots \cdots +\beta_{n}{\mathrm{REGU}}_{i,t}I\left(q_{i,t}>\gamma_{n}\right)+{\theta X}_{i,t}+u_{i}+e_{i,t}$$
where i represents the city; t represents the year; ${\mathrm{GTFP}}_{i,t}$ is the green total factor productivity, reflecting the level of urban economic activity; ${\mathrm{REGU}}_{i,t}$ represents the environmental regulation intensity; $q_{i,t}$ is the threshold variable(the per capita GDP level); $X_{i,t}$ is a set of control variables; $e_{i,t}$ is a random disturbance item; $u_{i}$ is an individual effect; I(·) is an indicative function. The value of the function is 1 when the conditions in the brackets are satisfied, and 0 otherwise. γ is the threshold value to be estimated; $\beta_{1}$~$\beta_{n}$ represents the elastic coefficient of the GTFP in different sections of the threshold variable, and indicates the existence of the threshold effect by determining whether the estimated value or sign of $\beta_{1}$~$\beta_{n}$ shows a significant difference.

### 2.2. Variable Selection

#### 2.2.1. Measurement of GTFP

Because the price information of resources and environmental factors cannot be obtained, the traditional total factor productivity measurement cannot calculate the productivity under resource and environmental constraints. Although the productivity index based on the traditional distance function does not require price information, it cannot calculate the total factor productivity in the presence of “bad” output (such as SO_2_ emissions). Chung et al. [7] first proposed the Malmquist–Luenberger (ML) index based on the directional distance function, which can measure the total factor productivity when there is a “bad” output. The existing research divides the data envelopment analysis (DEA) model of measuring efficiency into radial and angular, radial and non-angular, non-radial and angular, and non-radial and non-angular. “Radial” means that the input or output is required to change in the same proportion when evaluating efficiency, while “angled” means that when evaluating efficiency, it is necessary to make an input-based (assuming that output remains unchanged) or output-based (assuming that inputs are changed) DEA selection. Because DEA has the advantage of not requiring the assumption of the function form and being able to decompose productivity, many documents in the past basically used radial and oriented DEA to calculate the directional distance function. However, radial and angular DEA have certain shortcomings; that is, when there is over-input or under-output, the radial DEA efficiency measure will overestimate the efficiency of the evaluation object. In addition, the perspective of DEA efficiency measurement ignores a certain aspect of input or output, and the calculated efficiency result is not accurate. In order to overcome these two shortcomings, non-radial, non-angular directional distance function-slack-based measure (SBM) began to be widely used. Based on the existing non-radial and non-angle directional distance function (SBM) and Malmquist–Luenberger (ML) index research, this article refers to the method of Haifeng et al. [40] and adds energy input, using the Malmquist–Luenberger index based on the non-radial SBM directional distance, and then calculating the dynamic changes of the GTFP of 163 cities in China from 2003 to 2016.

Each city is regarded as a decision-making unit (DMU) to construct the production frontier. It is assumed that each DMU uses N kinds of inputs $x=x_{1,}\;x_{N,}\in R_{N}^{+}$ and produces M kinds of expected outputs $y=y_{1,}\;y_{M,}\in R_{M}^{+}$ accompanied by a Type I unexpected output$\;b=b_{1,}\;b_{I,}\in R_{I}^{+}$. In each period t =1,···,T the production possibility set of the k =1,···,K city was$\;\left(x^{k,t},y^{k,t},b^{k,t}\right)$. Data envelopment analysis (DEA) is used in the modeling as follows:(3)$$P^{t}\left(X^{t}\right)=\{\left(y^{t},b^{t}\right):\sum_{k=1}^{k}\lambda_{k}^{t}y_{\mathrm{km}}^{t}\geq y_{\mathrm{km}}^{t},\forall m;\sum_{k=1}^{k}\lambda_{k}^{t}b_{\mathrm{ki}}^{t}=y_{\mathrm{ki}}^{t},\forall i;\sum_{k=1}^{k}\lambda_{k}^{t}x_{\mathrm{kn}}^{t}\leq y_{\mathrm{kn}}^{t},\forall n;\sum_{k=1}^{k}\lambda_{k}^{t}=1,y_{k}^{t}\geq 0,\forall k;\}$$

This represents the weight of each cross-sectional observation value. $\sum_{k=1}^{k}\lambda_{k}^{t}=1,y_{k}^{t}\geq 0,\forall k$ means that the production technology has a variable return to scale.

Each city is considered to be a DMU to construct the productive frontier. The production possibility set of each DMU is expressed as (x, y, b). Because the directional distance function not only satisfies the relevant properties of the production possibility set but also reflects the directional properties shown by the output in the production process, the directional distance function is expressed by the following formula:(4)$$\vec{D_{0}}\left(x,\;y,\;b;g\right)=\sup\left\{\beta :\left(y,b\right)+\beta g\epsilon P\left(x\right)\right\}$$
where g = (gy, gb) is the direction vector, which is used to indicate the direction of output expansion. According to DEA, to solve the directional distance function, the following linear programming equation can be obtained:(5)$$\vec{D_{0}}\left(x^{t,k},y^{t,k},b^{t,k};y^{t,k},-b^{t,k}\right)=\mathrm{Max}\beta$$
(6)$$\sum_{k=1}^{k}\lambda_{k}^{t}y_{\mathrm{km}}^{t}\geq \left(1+\beta \right)y_{\mathrm{km}}^{t},m=1,2,\cdots ,M$$
(7)$$s.t.\overset{k}{\underset{k=1}{\sum }}\lambda_{k}^{t}b_{\mathrm{ki}}^{t}=\left(1-\beta \right)b_{\mathrm{ki}}^{t},i=1,2,\cdots ,I$$
(8)$$\overset{k}{\underset{k=1}{\sum }}\lambda_{k}^{t}b_{\mathrm{kn}}^{t}\leq x_{\mathrm{kn}}^{t},n=1,2,\cdot \cdot \cdot ,N$$

The ML productivity index is used to express GTFP based on environmental factors. The ML index from period t to period t+1 based on outputs is:(9)$${\mathrm{ML}}_{t}^{t+1}={\left\{\frac{\left[1+\vec{D_{0}^{t}}\left(x^{t},y^{t},b^{t};g^{t+1}\right)\right]}{\left[1+\vec{D_{0}^{t}}\left(x^{t+1},y^{t+1},b^{t+1};g^{t+1}\right)\right]}\times \frac{\left[1+\vec{D_{0}^{t+1}}\left(x^{t},y^{t},b^{t};g^{t+1}\right)\right]}{\left[1+\vec{D_{0}^{t+1}}\left(x^{t+1},y^{t+1},b^{t+1};g^{t+1}\right)\right]}\right\}}^{\frac{1}{2}}\;$$

The ML index can be further decomposed into a technological progress index (GTC) and a technological efficiency change index (GTEC). The GTC is mainly a measure of the shift of the production possibility boundary caused by technological progress, while GTEC is considered a measure of policy and system improvements.
(10)ML=GTEC×GTC

The expression of GTEC is:(11)$${\mathrm{GTEC}}_{t}^{t+1}=\frac{1+\vec{D_{0}}\left(x^{t},y^{t},b^{t};g^{t}\right)}{1+\vec{D_{0}}\left(x^{t+1},y^{t+1},b^{t+1};g^{t+1}\right)}$$

The expression of GTC is:(12)$${\mathrm{GTC}}_{t}^{t+1}={\left\{\frac{\left[1+\vec{D_{0}^{t}}\left(x^{t},y^{t},b^{t};g^{t}\right)\right]}{\left[1+\vec{D_{0}^{t}}\left(x^{t},y^{t},b^{t};g^{t}\right)\right]}\times \frac{\left[1+\vec{D_{0}^{t+1}}\left(x^{t+1},y^{t+1},b^{t+1};g^{t+1}\right)\right]}{\left[1+\vec{D_{0}^{t}}\left(x^{t+1},y^{t+1},b^{t+1};g^{t+1}\right)\right]}\right\}}^{\frac{1}{2}}$$

For the input indicators, we choose input variables such as human input, capital input, and energy consumption input [41,42,43]. For the output indicators, both the maximization of expected outputs (e.g., economic development) and undesired outputs (e.g., environmental pollution) are selected [8,44]. As a constraint on economic development, we choose the economic output as the expected output index and the environmental pollution index as the undesired output index. The research data come from the 2003–2016 “City Statistical Yearbook”.

ML > 1 indicates the growth of GTFP from period t to t + 1; ML < 1 indicates the decline of GTFP from period t to t + 1; ML = 1 indicates that GTFP is in a stable state. GTFP is obtained through the ML index. Specifically, the ML index is the growth rate of GTFP, which is a dynamic indicator. For example, Beijing’s ML index in 2006 was 1.0791, which means that Beijing’s GTFP in 2006 was 1.0791 times that of 2005, that is, an increase of 7.91% over 2005, and the same is true for the decomposition factors. Since the GTFP growth rate (ML index) and the decomposition items calculated by the ML index model are dynamic chain growth indicators, they reflect the improvement of the chain. In order to reasonably reflect the quality of economic growth in the current year, this article uses 2003 as the base period to convert the chain growth rate index of GTFP into a fixed rate improvement index, that is, assume that the environmental total factor productivity GTFP in 2003 is 1, and the GTFP growth rate multiply the index, that is, the GTFP in 2004 is the GTFP in 2003 multiplied by the ML index in 2004, the GTFP in 2005 is the GTFP in 2004 multiplied by the ML index in 2005, and so on, the GTFP of the corresponding year is obtained for the calculation of the empirical analysis.

#### 2.2.2. Environmental Regulation Intensity

The methods currently used to measure environmental regulation intensity are not uniform but mainly fall into the following two categories. The first is to use different pollutant emission densities to express the environmental regulation intensity [45,46,47]. The second is the use of environmental regulation policies to quantify the environmental regulation intensity [48]. Because the above indicators are relatively simple and are insufficient to represent the effectiveness of environmental regulations, a comprehensive index method is used to construct a comprehensive measurement system. Five individual items are selected: industrial SO_2_ removal rate (SO_2_), smoke and dust removal rate (DUST), comprehensive industrial solid waste utilization rate (SOLID), domestic sewage treatment rate (SEWAGE), and domestic waste harmless treatment rate (WASTE). The indicators used to measure environmental regulations are determined as follows.

The first step is to standardize the raw data:(13)$$P_{\mathrm{ij}}^{\prime \prime }=\frac{X_{\mathrm{ij}}-\min\left(X_{\mathrm{ij}}\right)}{\max\left(X_{\mathrm{ij}}\right)-\min\left(X_{\mathrm{ij}}\right)}$$
where $X_{\mathrm{ij}}$ represents the value of the j environmental pollution index of the i city.

The second step is to perform a coordinate translation on the standardized data:(14)$$P_{\mathrm{ij}}^{\prime }=1+P_{\mathrm{ij}}^{\prime \prime }$$

The third step is to calculate the proportion of the j environmental pollution index in the i city:(15)$$P_{\mathrm{ij}}=P_{\mathrm{ij}}^{\prime }/\sum_{i=1}^{m}P_{\mathrm{ij}}^{\prime }$$

The fourth step is to calculate the entropy and coefficient of variation of the j environmental pollution index:(16)$$e_{j}=\left(\frac{1}{\ln m}\right)\sum_{i=1}^{m}P_{\mathrm{ij}}\ln\left(P_{\mathrm{ij}}\right)$$
(17)$$g_{j}=1-e_{j}$$

The fifth step is to calculate the weight of the j environmental pollution index in the comprehensive evaluation:(18)$$W_{j=}g_{j}/\sum_{j=1}^{n}g_{j}$$

The sixth step is to calculate the comprehensive environmental pollution index:(19)$${\mathrm{REGU}}_{i}=\sum_{j=1}^{n}W_{j}P_{\mathrm{ij}}$$
where,${\;\mathrm{REGU}}_{i}$ represents the comprehensive environmental pollution index of city i. The larger the value of${\;\mathrm{REGU}}_{i}\;$, the higher the degree of environmental pollution in city i.

#### 2.2.3. Other Variables

The core explanatory variables in this study are environmental regulation intensity and five different environmental indicators. The industrial SO_2_ removal rate, the smoke and dust removal rate, the comprehensive industrial solid waste utilization rate, the domestic sewage treatment rate, and the domestic waste harmless treatment rate are selected, and the entropy value is obtained through standardization to finally obtain the environmental regulation intensity of municipal units.

Per capita GDP is selected as the threshold variable reflecting the urban economic development level [49,50]. Referring to existing studies, industrial structure (IS), government behavior, foreign direct investment (FDI), infrastructure, innovation capability, and technology level are also identified as variables [13]. The data in this paper are obtained from the 2003–2016 China urban statistical yearbook. The specific details of the variables are shown in Table 1.

### 2.3. Data Sources

The main research object of this paper is prefecture-level cities and above cities in my country. Due to the continuity of data and the availability of variables, the final sample cities selected are 163 prefecture-level cities and above, and the time span is 2003–2016—a total of 2282 sample values for 14 years. Among them, 41 are in eastern China, 91 are in central China, and 31 are in western China. These include both national-level important urban agglomerations and cities in underdeveloped regions. The selected variables are representative. The data in this article are mainly derived from the “China City Statistical Yearbook” (2004–2017) and “China Regional Economic Statistical Yearbook” (2004–2017).

## 3. Empirical Results

### 3.1. Descriptive Analysis

It can be seen from the above that GTFP is obtained through the ML index. Specifically, the ML index is the growth rate of GTFP, which is a dynamic indicator. Since the GTFP growth rate (ML index) and the decomposition items calculated by the ML index model are dynamic chain growth indicators, they reflect the improvement of the chain. In order to reasonably reflect the quality of economic growth in the current year, this paper uses 2003 as the base period to convert the chain growth rate index of GTFP into a fixed rate improvement index, that is, assume that the environmental total factor productivity GTFP in 2003 is 1, and the GTFP growth rate is multiplied by the index, that is, the GTFP in 2004 is the GTFP in 2003 multiplied by the ML index in 2004, the GTFP in 2005 is the GTFP in 2004 multiplied by the ML index in 2005, and so on to get the GTFP in the corresponding year.

From 2003 to 2016, the GTFP and ML indexes of China are shown in Figure 1. When the ML index is greater than 1 (red dotted line), it means that the GTFP of the year has increased compared to the previous year, and vice versa, it has decreased compared with the previous year, which also directly affects the trend of GTFP. It can be seen from the figure that from 2003 to 2006, GTFP showed a downward trend. Since 2007, although there has been a slight decline in the middle, GTFP has shown an upward trend as a whole.

The reasons for this situation are as follows: first, as the Chinese government attaches more importance to environmental protection year by year; second, when the global financial crisis began in 2007, the Chinese government adopted a series of short-term economic stimulus policies to reduce the impact of the financial crisis. However, as the effects of the short-term economic stimulus policies adopted to ease the economic crisis continue to weaken, China’s economic growth momentum has not yet been effectively converted, and economic growth has been weak. Insufficient endogenous motivation for enterprises to invest in technological innovation to promote technological progress, coupled with the impact of the global financial crisis in 2008, resulted in a certain decline from 2009 to 2011. However, urban enterprises are also facing strong constraints on energy conservation and emission reduction, thus maintaining the overall growth trend of GTFP.

Figure 2 shows the changes in the emission reduction effects of five pollutants in 163 cities from 2003 to 2016. It can be seen that the industrial SO_2_ removal rate is much lower than other pollutant emission reduction rates, but the SO_2_ removal rate has increased the most significantly, and the growth rate is the fastest. Among the five indicators, the soot removal rate (DUST) has always been maintained at a leading position, but the growth rate is relatively low. The comprehensive utilization rate of industrial solid waste (SOLID), the treatment rate of domestic sewage (SEWAGE), and the rate of harmless treatment of domestic garbage (GARBAGE) have always maintained a stable growth state. Regarding the environmental regulation intensity index (REGU), although there was a certain degree of decline in 2006, 2011, and 2015, it still showed a significant growth state as a whole. It can be seen from the government’s emphasis on environmental regulation.

### 3.2. Threshold Model

The results of a threshold effect test with per capita GDP as the threshold variable are shown in Table 2. To test the impact of different environmental indicators on GTFP, five indicators were used as explanatory variables in the threshold model to quantify the environmental regulation intensity. The corresponding P values and confidence intervals were obtained by bootstrap sampling 300 times [39]. It can be seen from Table 2 that the threshold test results of each model rejected the single and triple thresholds, while the double threshold passed the threshold effect test, indicating that the double threshold was suitable for this study. The threshold values of environmental regulation intensity and four indicators (industrial SO_2_ removal rate, smoke and dust removal rate, comprehensive industrial solid waste utilization rate, and domestic sewage treatment rate) were 12,873 and 55,447. Therefore, the urban per capita GDP was divided into three stages of low level (GDP ≤ 12,873), medium level (12,873 < GDP ≤ 55,447), and high level (GDP > 55,447). The two threshold values of the domestic waste harmless treatment rate were 11,032 and 55,447, respectively, which means that when the explanatory variable was the domestic waste harmless treatment rate, the threshold value divided the urban per capita GDP into three stages of low level (GDP ≤ 11,032), medium level (11,032 < GDP ≤ 55,447), and high level (GDP > 55,447).

Table 3 shows the regression results of the threshold model. For each model, a threshold model was used to estimate the parameters. As shown in the first column of Table 3, when per capita GDP was low (GDP ≤ 12,873), the environmental regulation intensity had a significant positive impact on GTFP at the 1% level. When per capita GDP was at a medium level (12,873 < GDP ≤ 55,447) and high level (GDP > 55,447), the impact of environmental regulation intensity on GTFP was initially negative, but then became positive over time, although these two effects were not significant. When the explanatory variables were the industrial SO_2_ removal rate, smoke and dust removal rate, comprehensive industrial solid waste utilization rate, and domestic sewage treatment rate, the three stages of per capita GDP were at a low level (GDP ≤ 12,873), medium level (12,873 < GDP ≤ 55,447), or high level (GDP > 55,447). In each stage, the effects of these four indicators on GTFP differed.

As shown in column (2) of Table 3, the impact of the industrial SO_2_ removal rate on GTFP was positive at a low economic development level (GDP ≤ 12,873) and was significant at the 1% level, which was basically the same as the impact of the environmental regulation intensity as an explanatory variable. At the medium level of per capita GDP (12,873 < GDP ≤ 55,447), although the industrial SO_2_ removal rate still had a positive effect on GTFP, the influence coefficient decreased and was not significant. When the per capita GDP increased to a high level (GDP > 55,447), the positive impact began to gradually recover.

As shown in column (3) of Table 3, the change in the impact of the smoke and dust removal rate on GTFP was very different from that of the industrial SO_2_ removal rate. Under the low economic development level (GDP ≤ 12,873), the impact on GTFP was positive but not significant. Under the medium level of per capita GDP (12,873 < GDP ≤ 55,447), the impact on GTFP was negative and significant at the 1% level. When the per capita GDP was high (GDP > 55,447), the impact on GTFP was negative but not significant.

As shown in column (4) of Table 3, the impact of the comprehensive industrial solid waste utilization rate was similar to that of the smoke and dust removal rate. The impact on GTFP was positive and significant at low levels of economic development (GDP ≤ 12,873). At the medium level of per capita GDP (12,873 < GDP ≤ 55,447), the comprehensive industrial solid waste utilization rate had a negative effect on the GTFP that was significant at the 1% level. When it reached a high economic level (GDP > 55,447), the coefficient remained significantly negative.

As shown in column (5) of Table 3, the domestic sewage treatment rate had similar effects on the GTFP and the environmental regulation intensity. When the per capita GDP was low (GDP ≤ 12,873), the domestic sewage treatment rate had a positive effect on the GTFP that was significant at the 1% level, with an R2 value of 0.238. When per capita GDP was at a medium level (12,873 < GDP ≤ 55,447) and a high level (GDP > 55,447), the impact of the domestic sewage treatment rate on GTFP was initially negative, but became positive over time.

As shown in column (6) of Table 3, when the domestic waste harmless treatment rate was used as the explanatory variable, the threshold value was slightly different from the previous models. Under the threshold values of 11,032 and 55,447, the per capita GDP was divided into three intervals, namely a low economic development level (GDP ≤ 11,032), medium level (11,032 < GDP ≤ 55,447), and high level (GDP > 55,447). When the economic development level was low (GDP ≤ 11,032), the impact of the domestic waste harmless treatment rate on GTFP was positive and significant at the 1% level. When the economic development level reached the medium stage (11,032 < GDP ≤ 55,447), the impact on GTFP changed from positive to negative, and was significant at the 10% level. When the economic level reaches a high level (GDP > 55,447), the impact is not significant.

For the other control variables, the impact of industrial structure on GTFP was always significantly positive, i.e., the greater the proportion of the tertiary industry, the higher the urban GTFP. In contrast, the impact of power consumption per unit GDP on GTFP was always significantly negative. The smaller the power consumption per unit GDP, the better the technology available for energy conservation and emission reduction, and the GTFP will increase accordingly.

In conclusion, when the explanatory variables were the domestic sewage treatment rate and domestic waste harmless treatment rate, there was a significant positive impact on GTFP at low levels of economic development. With an improvement in the economic development level, the impact changed from positive to negative, but at high levels of economic development, the impact became positive again, although the influence coefficient and level of significance decreased. This was very similar to the impact of environmental regulation intensity on GTFP. The industrial SO_2_ removal rate had a positive impact on GTFP, and was significant in most stages of economic development. The effects of the smoke and dust removal rate and solid waste comprehensive utilization rate on GTFP were similar. At medium and high levels of economic development, these two indicators had a significant negative impact on GTFP. Therefore, based on the economic development level of an individual city, attention should be given to the extent of these five environmental indicators in the formulation of environmental regulation policies, thus ensuring a stable and sustainable improvement of GTFP.

### 3.3. Robustness Testing

#### 3.3.1. The Robustness of Threshold Variables

GDP is an important indicator of economic growth. Data quality has a large influence on research outcomes. Different statistical calibers, over-classification of sectors, and inaccurate price indices may affect the quality of GDP data [51]. Researchers are therefore committed to finding an objective measure of the economic development level based on alternative indicators. Night light data observed using satellite technology are considered a good alternative and the correlation between regional light brightness and GDP has been generally proven [52,53,54,55]. In this paper, NL is used to represent Night light data as a symble.

Nighttime light data are used to measure and reflect economic development, mainly through the use of Defense Meteorological Satellite Program/Operational Line Scanner (DMSP/OLS) night light data. It can be seen from Table 4 that in the model with the environmental regulation as the explanatory variable. The nighttime lighting was divided into two stages: low level (NL ≤ 2.670) and high level (NL > 2.670). The threshold value of the industrial SO_2_ removal rate was the same as that of the environmental regulation intensity, with a threshold value of 2.670. The threshold value of the smoke and dust removal rate was slightly higher at 2.732. The comprehensive industrial solid waste utilization rate had a double threshold, with threshold values of 1.845 and 8.204, respectively. There was no threshold effect in the models of the domestic sewage treatment rate and domestic waste harmless treatment rate.

Table 5 shows the impacts of the various environmental indicators on GTFP when nighttime light was used as the threshold variable (because there was no threshold effect in the models of domestic sewage treatment rate and domestic waste harmless treatment rate, the threshold regression process was not applied). It can be seen from the first column of Table 5 that the impact of environmental regulation intensity on GTFP changed with an increase in night light levels. At the low level of night lighting (NL ≤2.670), the impact of environmental regulation intensity on GTFP was positive, whereas when the nighttime lighting level was high (NL > 2.670), the environmental regulation intensity had a negative impact on GTFP, but neither value was significant. These changes were similar to the changes in the effect of environmental regulation intensity on GTFP under the threshold variable of per capita GDP.

The various indicators of environmental regulation were decomposed for use as explanatory variables. When the night lighting level was taken as the threshold variable, the industrial SO_2_ removal rate (column 2 of Table 5) and the comprehensive industrial solid waste utilization rate (column 4 of Table 5) were found to have a significant positive effect on GTFP. In contrast, the smoke and dust removal rate (column 3 of Table 5) had a negative impact, which was basically consistent with the per capita GDP results. It can therefore be concluded that our results were robust.

#### 3.3.2. Hysteresis Test

Because there may be a time lag in the impact of environmental regulation [28,56], a lag period in the impact of environmental regulation was introduced as an explanatory variable for the regression. Table 6 shows the test results of the threshold effect under a lag in the impact of environmental regulation.

It can be seen from Table 6 that in the model with the environmental regulation intensity obtained by the entropy weight method as an explanatory variable, the threshold test results were double thresholds, with threshold values of 11,158 and 1965. According to these two thresholds, per capita GDP could be divided into low level (GDP ≤ 11,158), medium level (11,158 < GDP ≤ 1965), and high level (GDP > 1965). The threshold values of the industrial SO_2_ removal rate and domestic sewage treatment rate were the same at 12,140 and 55,089. Based on these values, three intervals were established, with a low level (GDP ≤ 12,140), medium level (12,140 < GDP ≤ 55,089), and high level (GDP > 55,089).

Table 7 shows the impact of an environmental regulation lag on GTFP when per capita GDP was the threshold variable. It can be seen from Table 7 that the impact of the environmental regulation lag on GTFP was stronger than the immediate impact. For environmental regulation intensity, under the low economic development level (GDP ≤ 11,158) and the medium level (11,158 < GDP ≤ 19,656), the lag in the impact of environmental regulation had a significant positive impact on GTFP, with the coefficient and significance being greater than the corresponding values for the immediate impact of environmental regulation. In addition, the impact of a lag in the industrial SO_2_ removal rate, smoke and dust removal rate, comprehensive industrial solid waste utilization rate, domestic sewage treatment rate, and domestic waste harmless treatment rate on GTFP were basically the same as the economic development level changed, which further confirmed the robustness of the model.

## 4. Discussions and Conclusions

Based on the non-radial SBM directional distance function and the ML productivity index, the GTFP of 163 cities in China was measured. Per capita GDP was used as the threshold variable to measure the economic development level of each city, and five environmental indicators were also selected: the industrial SO_2_ removal rate, smoke and dust removal rate, comprehensive industrial solid waste utilization rate, domestic sewage treatment rate, and domestic waste harmless treatment rate. An empirical analysis was conducted of the relationship between the environmental indicators that indicated the environmental regulation intensity and GTFP.

In both developed and developing countries, environmental regulations are increasingly recognized as an important way to improve GTFP [57]. It is generally thought that whether an environmental quality improvement and total factor productivity improvement can achieve a “win-win” depends on the size of both the “cost loss” and “revenue compensation” [36]. The government has imposed strict environmental regulations to force companies to reduce pollution emissions. While improving the environment, these regulations also cause a “cost loss” for companies, which has a negative impact on GTFP. Specifically, environmental regulations can increase corporate costs and affect the upgrading of corporate technology, which in turn enables companies to increase output driven by the goal of profit maximization, causing more pollution emissions, and negatively impacting GTFP; there are also some companies moving to avoid the high-strength environmental regulations of the original city, which also caused the decline of the original urban total factor productivity [58]. For example, Dufour et al. analyzed the manufacturing data of Quebec, Canada, and found that the increase in the intensity of government environmental regulations has led to a decline in the growth rate of total factor productivity [59]. “Cost loss” is divided into two parts: “direct cost loss” and “indirect cost loss”. First, the implementation of environmental regulatory policies requires costs such as manpower and material resources, which are “direct cost losses” [18]; second, after the promulgation of environmental regulatory policies, a large number of funds will be invested in environmental foundations. The construction of facilities has squeezed out potential efficiency investment or innovation to a certain extent. This part of the squeeze of production investment due to investment in environmental pollution control is an “indirect cost loss” [15,16]. However, the implementation of environmental regulation policies has promoted a more rational allocation of resources, and the innovative application of corporate green technologies has stimulated a “revenue compensation” effect within companies, which has a positive impact on total factor productivity [60,61]. “Revenue compensation” is also divided into two parts: “direct revenue compensation” and “indirect revenue compensation”. First of all, the implementation of environmental regulations has brought about the improvement of environmental quality and reduced the economic and health losses caused by environmental problems. This part of the reduction in losses caused by the improvement of environmental quality belongs to “direct benefit compensation” [25]. Secondly, the implementation of environmental regulation policies has promoted the rapid development of the environmental protection industry, partly because the economic benefits brought by the development of the environmental protection industry belong to “indirect revenue compensation”. Furthermore, the implementation of environmental regulations has promoted technological progress in various enterprises, which in turn promoted the improvement of green total element projections. Specifically, the increase in the intensity of environmental regulations can enable companies to increase research and development (R&D) investment and carry out innovations in factor input, energy consumption, energy conservation, and emission reduction, which can further improve the competitiveness of the company and increase the output of the company and make up for the cost of environmental governance. The decline in corporate profits caused by the increase has continuously promoted the increase in GTFP. This part of the economic benefits brought about by technological progress also belongs to “indirect revenue supplement” [25,27].

When the “cost loss” is greater than the “revenue compensation”, the impact of environmental regulations on GTFP is negative; when the “cost loss” is less than the “revenue compensation”, the impact of environmental regulations on GTFP is positive. Based on the above analysis, the process of environmental regulation on GTFP is shown in Figure 3.

The results of this paper show that: for the intensity of environmental regulations, when the level of urban economic development is at a lower stage, the intensity of environmental regulations has a positive and significant impact on GTFP, which is consistent with the fact that “cost loss” is less than “revenue compensation” The situation also confirms the viewpoint of the “Porter Hypothesis”. That is to say, reasonable environmental regulations can promote technological innovation by enterprises in cities, thereby increasing GTFP [62,63,64,65,66,67,68]. With the improvement of the economic level, the impact of environmental regulations on GTFP undergoes a process from positive to negative and then to positive, but the impact is not significant.

For the five pollutant emission reduction indicators, when the city’s economic level is at a low level, the five pollutant emission reduction effects have a positive impact on the improvement of GTFP; that is, the “cost loss” is less than the “revenue compensation”. Among the five pollutants, the soot removal rate has no significant impact on the GTFP, and the other four pollutants can significantly increase the GTFP. Industrial SO_2_ removal rate has the greatest positive impact on GTFP.

When the urban economic development level was in the middle stage, the four indicators of smoke and dust removal rate, comprehensive industrial solid waste utilization rate, domestic sewage treatment rate, and domestic waste harmless treatment rate not only improved the GTFP but also affected economic development. The negative impact of the smoke and dust removal rate was the most significant impact. The negative impact was caused by the actions required to undertake these activities consuming manpower and material resources. According to the “following cost theory”, companies in cities must use cleaner raw materials or invest in more resources to control pollution to meet environmental standards. This will inevitably squeeze investment and funds originally allocated for research and development (R&D), reduce corporate profit margins, and be detrimental to corporate technological progress, thereby affecting the improvement of urban GTFP [69], i.e., the “cost loss” is greater than the “profit supplement”. In the past two years, many researchers have reached similar conclusions. For example, green innovation data in 30 provinces and regions in China have been analyzed and it was found that neither regulatory policies nor government subsidy policies have been able to promote green development [70].

When the economic development level was high, the industrial SO_2_ removal rate was the most useful indicator of the improvement of GTFP, and had a positive impact on the impact of GTFP. The comprehensive industrial solid waste utilization rate also had a positive impact on GTFP. Other environmental indicators had a significant negative impact. This is consistent with the belief that “the impact of environmental regulations on GTFP cannot be determined”. The impact of environmental regulations on GTFP is not only affected by the threshold effect of various variables [29,30]; different types of environmental regulations will have different effects on GTFP [71], and some of these differences may be regional [57].

Based on the above main conclusions, the following policy enlightenments are drawn.

As the Chinese economy shifts from high-speed growth to high-quality growth, it is necessary to appropriately strengthen environmental regulations, but it is not suitable for a one-size-fits-all approach. The government has played a primary role in the formulation and implementation of environmental regulations. On the one hand, the government should combine the local economic level and other characteristics when formulating environmental regulation policies to give full play to the positive effect of environmental regulation on green total factor productivity. Local governments at all levels should keep the intensity of environmental regulations within a reasonable range according to the city’s own characteristics and economic development level. The government’s regulatory tools and regulatory efforts should take into account local characteristics and formulate differentiated environmental regulatory policies based on local conditions. On the other hand, the government should appropriately improve relevant laws and regulations to strengthen the supervision and punishment of environmental pollution. At the same time, cities should adjust environmental regulations and policies according to the actual situation of local pollutant discharge to maintain flexibility and effectiveness. For example, the industrial SO_2_ removal rate has always shown a positive effect on green total factor productivity in various urban economic development stages. Cities can choose to formulate strict environmental regulation policies on SO_2_ emissions to help urban green total factor productivity increase.

Industrial structure and electricity consumption per unit of GDP have a significant impact on green total factor productivity. Therefore, in order to increase green total factor productivity, cities should increase the development of the tertiary industry and the use of clean energy. China must increase the intensity and pace of energy structure adjustment, increase the development and utilization of clean energy, and reduce the proportion of coal in energy consumption. For example, the Chinese government can encourage the supply of clean energy such as wind and solar energy through price subsidies and policy preferences; the Chinese government can also reduce irrational behaviors through internalized environmental external costs of energy use such as resource taxes and environmental taxes. Encourage clean energy consumption. Ultimately speed up the promotion and use of cleaner production technologies to reduce pollution emissions.

## Figures and Tables

**Figure 1 ijerph-18-04828-f001:**
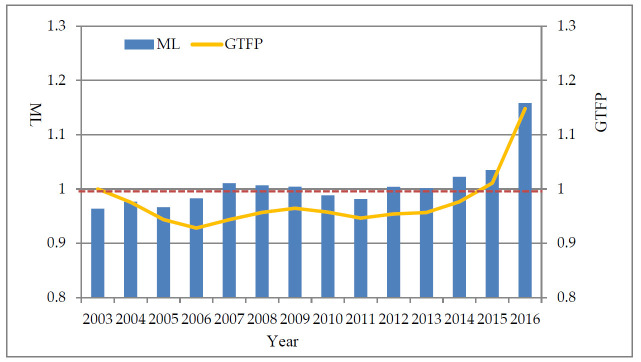
MLMalmquist–Luenberger (ML) index and Green Total Factor Productivity (GTFP) from 2003 to 2016.

**Figure 2 ijerph-18-04828-f002:**
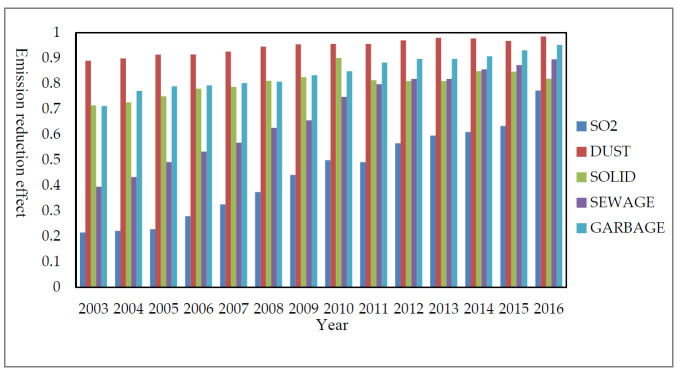
Changes in the five environmental indicators from 2003 to 2016.

**Figure 3 ijerph-18-04828-f003:**
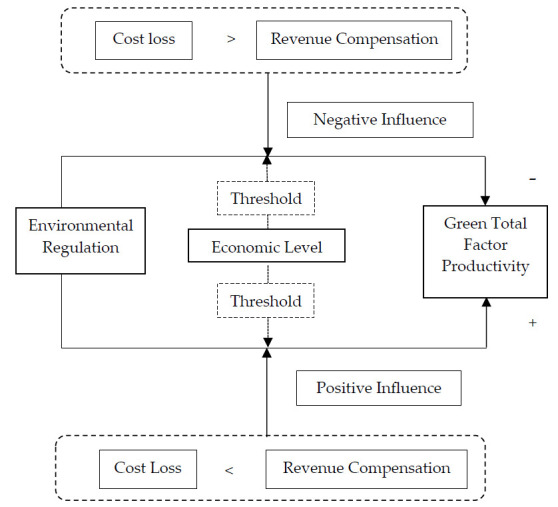
The mechanism of environmental regulation on GTFP.

**Table 1 ijerph-18-04828-t001:** Variable definition.

Classification	Name	Interpretation
Explained variable	Green total factor productivity (GTFP)	Malmquist–Luenberger exponent calculation based on non-radial -slack-based measure (SBM) directional distance
Explanatory variables	Industrial SO2 removal rate (SO_2_)	The intensity of environmental regulation is calculated by entropy weight method through the five single indexes of industrial SO_2_ removal rate (SO_2_), smoke and dust removal rate (dust), comprehensive utilization rate of industrial solid waste(solid), domestic sewage treatment rate (sewage) and harmless treatment rate of domestic garbage(garbage)
	Smoke and dust removal rate (DUST)	
	Comprehensive utilization rate of industrial solid waste (SOLID)	
	Domestic sewage treatment rate (SEWAGE)	
	Harmless treatment rate of domestic garbage (GARBAGE)	
	Strength of environmental regulations (REGU)	
Threshold variable	Regional economy (GDP)	GDP per capita
Control variable	Industrial structure (IS)	Added value of tertiary industry/added value of secondary industry
	Open to the outside world (FDI)	Industrial output value of foreign-invested enterprises/Gross regional product
	Government Action (GOV)	The ratio of education and technology expenditure to the regional GDP
	Infrastructure (ROD)	Urban road area per capita
	Innovation capacity (RD)	Number of patents granted
	Technology level (TECH)	Power consumption per unit GDP

**Table 2 ijerph-18-04828-t002:** Results of the threshold effect test: GDP as the threshold.

Environmental Regulation Category	Model	Threshold	F-Statistic (F)	*p*-Value (*p*)	Bootstrap (BS)
REGU	Single threshold	12,873	125.74 ***	0	300
	Double threshold	12,873	41.89 **	0.0167	300
		55,447			
	Three thresholds	19,824	44.78	0.4367	300
SO_2_	Single threshold	55,447	63.23 ***	0	300
	Double threshold	12,873	50.03 ***	0.01	300
		55,447			
	Three thresholds	35,333	11	0.6833	300
DUST	Single threshold	12,873	119.26 ***	0	300
	Double threshold	12,873	35.55 ***	0.01	300
		55,447			
	Three thresholds	19,824	29.59	0.267	300
SOLID	Single threshold	12,873	104.01 ***	0	300
	Double threshold	12,873	51.25 ***	0. 0067	300
		55,447			
	Three thresholds	19,824	30.55	0.3367	300
SEWAGE	Single threshold	12,873	98.65 ***	0	300
	Double threshold	12,873	52.41 ***	0.0033	300
		55,447			
	Three thresholds	17,594	13.21	0.55	300
GARBAGE	Single threshold	11,032	113.26 ***	0	300
	Double threshold	11,032	41.25 **	0.0167	300
		55,447			
	Three thresholds	19,824	28.23	0.4567	300

Note: *, **, and *** represent that the estimated coefficient is significant at the 10%, 5%, and 1% confidence levels, respectively.

**Table 3 ijerph-18-04828-t003:** Parameter results of the model: GDP is the threshold.

Variable	GDP is the Threshold					
	REGU	SO_2_	DUST	SOLID	SEWAGE	GARBAGE
GDP-1	0.280 ***	0.348 ***	0.016	0.105 *	0.238 ***	0.206 ***
	(0.100)	(0.096)	(0.074)	(0.059)	(0.091)	(0.074)
GDP-2	−0.025	0.002	−0.166 ***	−0.082 ***	−0.075 *	−0.06 *
	(0.077)	(0.046)	(0.072)	(0.020)	(0.046)	(0.035)
GDP-3	0.063	0.126 ***	−0.105	−0.001 ***	0.015	0.008
	(0.077)	(0.046)	(0.074)	(0.000)	(0.047)	(0.038)
IS	0.115 ***	0.11 **	0.114 ***	0.12 ***	0.115 ***	0.117 ***
	(0.042)	(0.044)	(0.041)	(0.041)	(0.041)	(0.042)
FDI	−0.061	−0.068	−0.054	−0.066	−0.051	−0.058
	(0.0645)	(0.066)	(0.065)	(0.066)	(0.065)	(0.066)
ROD	0.0011	0.0003	0.002	0.001	0.002	0.001
	(0.002)	(0.002)	(0.002)	(0.002)	(0.002)	(0.002)
GOV	0.624	−0.134	0.831	0.649	0.827	0.59
	(0.821)	(0.873)	(0.845)	(0.811)	(0.834)	(0.783)
RD	0.009515	0.005	0.0121	0.008	0.011	0.0108641
	(0.022)	(0.020)	(0.022)	(0.021)	(0.022)	(0.022)
TECH	−0.258 **	−0.232 *	−0.28 ***	−0.278 **	−0.267 **	−0.25 **
	(0.120)	(0.119)	(0.123)	(0.122)	(0.121)	(0.117)
Constant	0.874 ***	0.89 ***	1. 007 ***	0.923 ***	0.896 ***	0.906 ***
	(0.063)	(0.051)	(0.08)	(0.053)	(0.055)	(0.057)
Numbers	163	163	163	163	163	163
R-squared	0.13	0.11	0.13	0.13	0.13	0.13

Note: *, **, and *** represent that the estimated coefficient is significant at the 10%, 5%, and 1% confidence levels, respectively. The standard errors of the coefficients are marked in parentheses.

**Table 4 ijerph-18-04828-t004:** Threshold effect test results: NL (Nighttime light) is the threshold.

Environmental Regulation	Model	Threshold	F	P	BS
REGU	Single threshold	2.670	44.95 *	0.053	300
	Double threshold	1.845	40.55	0.137	300
		2.732			
SO_2_	Single threshold	6.834	61.99 ***	0	300
	Double threshold	2.670	28.13	0.2333	300
		6.834			
DUST	Single threshold	2.732	48.21 **	0.02	300
	Double threshold	1.845	40.2	0.13	300
		2.732			
SOLID	Single threshold	1.845	39.31 *	0.093	300
	Double threshold	1.845	39.92 *	0.083	300
		8.204			
	Three thresholds	0.428	33.94	0.383	300
SEWAGE	Single threshold	8.483	32.53	0.1833	300
GARBAGE	Single threshold	2.670	37.77	0.11	300

Note: *, **, and *** represent that the estimated coefficient is significant at the 10%, 5%, and 1% confidence levels, respectively.

**Table 5 ijerph-18-04828-t005:** Parameter results of the model: NL (Nighttime light) is the threshold.

Variable	NL (Nighttime Light) Is the Threshold			
	REGU	SO_2_	DUST	SOLID
NL-1	0.09	0.09 *	−0.027	0.044
	(0.077)	(0.05)	(0.070)	(0.055)
NL-2	−0.122	0.14 **	−0.182 **	0.110 ***
	(0.074)	(0.059)	(0.077)	(0.038)
NL-3				0.001 **
				(0.0003)
IS	0.14 ***	0.14 ***	0.127 **	0.121 ***
	(0.037)	(0.036)	(0.036)	(0.036)
FDI	0.039	−0.019	0.041	0.016
	(0.054)	(0.056)	(0.055)	(0.053)
ROD	0.002	0.001	0.002	0.002
	(0.002)	(0.002)	(0.002)	(0.002)
GOV	0.484	−1.19	−0.538	−0.766
	(0.936)	(0.808)	(0.886)	(0.84)
RD	−0.001	−0.017	−0.002	−0.008
	(0.018)	(0.015)	(0.017)	(0.015)
TECH	−0.27 **	−0.241 *	−0.291 **	−0.28 **
	(0.136)	(0.133)	(0.137)	(0.041)
Constant	0.896 ***	0.887 ***	0.994 ***	0.927 ***
	(0.056)	(0.046)	(0.075)	(0.055)
Numbers	163	163	163	163

Note: *, **, and *** represent that the estimated coefficient is significant at the 10%, 5%, and 1% confidence levels, respectively. The standard errors of the coefficients are marked in parentheses.

**Table 6 ijerph-18-04828-t006:** Test results of threshold effect under lagging environmental regulation.

Environmental Regulation	Model	Threshold	F	*p*	BS
REGU-1	Single threshold	12,140	89.61 ***	0	300
	Double threshold	11,158	47.13 **	0.0167	300
		19,656			
	Three thresholds	53,771	31.5	0.39	300
SO_2_-1	Single threshold	12,140	51.32 ***	0.0033	300
	Double threshold	12,140	44.48 ***	0.0067	300
		55,089			
	Three thresholds	90,261	12	0.6533	300
DUST-1	Single threshold	17,421	86.2 ***	0	300
	Double threshold	17,421	32.66 **	0.0233	300
		53,771			
	Three thresholds	11,158	26.54	0.2367	300
SOLID-1	Single threshold	16,892	80.96 ***	0	300
	Double threshold	16,892	30.07 *	0.0567	300
		61,177			
	Three thresholds	11,158	18.45	0.35	300
SEWAGE-1	Single threshold	12,140	66.11 ***	0	300
	Double threshold	12,140	49.26 **	0.0233	300
		55,089			
	Three thresholds	19,656	20.69	0.3867	300
GARBAGE-1	Single threshold	12,140	81.4 ***	0	300
	Double threshold	12,140	33.37 *	0.06	300
		53,771			
	Three thresholds	19,656	28.28	0.23	300

Note: *, **, and *** represent that the estimated coefficient is significant at the 10%, 5%, and 1% confidence levels, respectively.

**Table 7 ijerph-18-04828-t007:** Parameter results of the model under lagging environmental regulations.

Variable	Environmental Regulation Lags One Step Behind					
	REGU-1	SO_2_-1	DUST-1	SOLID-1	SWAEGE-1	GARBAGE-1
GDP-1	0.5137 ***	0.364 ***	0.05	0.140 ***	0.271 ***	0.153 **
	(0.123)	(0.124)	(0.0795)	(0.045)	(0.096)	(0.070)
GDP-2	0.228 ***	−0.061	−0.174 **	−0.005	−0.043	−0.074 **
	(0.072)	(0.070)	(0.078)	(0.005)	(0.045)	(0.034)
GDP-3	0.068	0.102 **	−0.114	0.067 ***	0.047	−0.013
	(0.067)	(0.044)	(0.080)	(0.020)	(0.043)	(0.036)
IS	0.113 **	0.115 **	0.113 ***	0.118 ***	0114 **	0.12 ***
	(0.044)	(0.013)	(0.043)	(0.043)	(0.044)	(0.045)
FDI	−0.033	−0.061	−0.047	−0.060	−0.055	−0.06
	(0.069)	(0.07)	(0.0696)	(0.070)	(0.071)	(0.07)
ROD	0.003	0.001	0.002	0.002	0.001	0.002
	(0.002)	(0.002)	(0.002)	(0.002)	(0.002)	(0.002)
GOV	1.396	0.341	0.922	0.711	0.676	0.790
	(0.845)	(0.831)	(0.293)	(0.847)	(0.772)	(0.779)
RD	0.022	0.009	0.0155	0.013	0.012	0.0131
	(0.022)	(0.02)	(0.021)	(0.021)	(0.021)	(0.021)
TECH	−0.428 ***	−0.301 **	−0.381 **	−0.365 **	−0.286 **	−0.310 **
	(0.155)	(0.13)	(0.150)	(0.14)	(0.125)	(0.128)
Constant	0.799 ***	0.883 ***	1.01 ***	0.86 ***	−0.286 **	0.911 ***
	(0.062)	(0.054)	(0.085)	(0.052)	(0.125)	(0.057)
Numbers	163	163	163	163	163	163

Note: *, **, and *** represent that the estimated coefficient is significant at the 10%, 5%, and 1% confidence levels, respectively. The standard errors of the coefficients are marked in parentheses.

## Data Availability

Data were selected from “China City Statistical Yearbook” (2004–2017) and “China Regional Economic Statistical Yearbook” (2004–2017) at https://www.cnki.net/.

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
