# Peer review of "Does Environmental Regulation Improve the Green Total Factor Productivity of Chinese Cities? A Threshold Effect Analysis Based on the Economic Development Level"

_ijerph, 2021, doi:10.3390/ijerph18094828_

Round 1
Reviewer 1 Report
This paper aims to investigate the impact of environmental regulation on urban green total factor productivity under different levels of urban economic development. Some major issues need to be addressed to make the manuscript more readable and scientific.
The structure of the manuscript needs to be re-organized. 1)The only two figures of the paper in "Data and methods" section should be moved to "Results" since the numbers in the figures are derived from the authors' calculation. "Data and methods" section should just involve the methodology, experiment design, and data source description. 2)There is no description of the basic information (e.g., geographical info, location in the country, basic economic condition, etc) of the selected 163 cities. Are there any criteria to do the selection? Are the cities representative? Do the data source cover the general situation nationally? 3)"Research design" section should be combined with "Data and Method" section.
Since the authors used a lot of equations (17 in total) and narratives to explain the methodology or analysis of the study, it's not that "decipherable". A schematic diagram depicting all the inputs, general process, and outputs of this study would be helpful for readers to understand the logic and the process flow of the study.
There are only two figures in the main text without labels of the x-axis, y-axis (secondary y-axis). So many results are presented in the tables making the findings not that intriguing for readers. Some of the results may be easier to be interpreted in bar figures or other informative figures.
The content of "Discussion" and "Conclusion" sections are repetitious. Combine them as "Conclusion and discussion" to make the content less long-winded.
Detailed comments:
- Line 50: the first present of "R&D" should be in full name with its abbreviation in parentheses.
- Some of the narratives are too wordy which repeats similar content. Such as in line 61-63: "For example, Zhou et al. [28] identified a nonlinear relationship between environmental regulation and GTFP." describes the same content as the previous sentence "There is a nonlinear relationship between environmental regulations and GTFP."
- Line 102: " Back..."?
- Line 103: what does “GML” stand for?
- In equation (5) and (6), what does "ML" represent? What is the full name?
Reviewer 2 Report
The reviewed article focuses on an interesting topic of Green Total Factor Productivity (GTFP), analyzing it through the prism of the Environmental Regulation and its impact in GTFP in Chinese Cities. In particular, the authors explored the impact of environmental regulation intensity on GTFP based on panel data for 163 prefecture-level cities from 2003 to 2016. Three aims of this study are properly formulated regarding the impact of environmental regulation on GTFP. In my opinion, the Paper is potentially interesting and valuable for International Journal of Environmental Research and Public Health readers.
Author Response
Special thanks to you for your good comments.Thanks for your good guidance and help.
Reviewer 3 Report
The authors study the relationship between per capita GDP and green total factor productivity under environmental regulation shaped by the SO2 removal rate, smoke and dust removal rate, solid waste utilization rate, domestic sewage treatment rate, and waste harmless treatment rate. The purpose is to identify the impact of these policy instruments on the stage of economic growth in China between 2003 and 2016, which, of course, is very interesting policy-wise. The paper is well-written with exhaustive literature review. It is original, interesting, and contains new results that advance the research field. It is thus very likely to be cited in the future. The studies of the relationship between environmental regulation and GTFP at the urban level are only few. And, the existing research on this relationship is inconclusive due to several reasons. This paper sort of fills this gap through data envelopment analysis.
Author Response
Special thinks to you for your good comments.Thank you for your guidance and help.
Reviewer 4 Report
Focusing on a panel of 163 cities located in China, this paper investigates the effects of environmental regulation on green total factor productivity (GTFP). The topic is rather interesting although the manuscript needs substantial revision before it can be published. Hopefully the comments below will help the authors when pursuing this objective.
Comments:
- Keywords: I would delete ‘GDP’ since it is too generic and not really reflecting the focus of the paper. I would add ‘China’ and ‘GML productivity index’ as key words.
- Line 31: Replace ‘arisen’ with ‘emerged’.
- Line 34: Add a reference to support this sentence.
- Lines 41-66: Please, condense the following text in a single sentence: ‘For example, 41 the ‘air pollution prevention and control law’ was revised in October 2018. Although the Chinese 42 government is actively looking for ways to improve GTFP’. Then, start a new paragraph using the text from lines 43-48. After that present the three hypothesis. Please, introduce the hypothesis in a consistent manner, e.g. using the same editorial style. At the moment two of them are presented in individual paragraphs while the first one is presented together with additional text.
- Line 51: Please, provide a reference for "pollution paradise hypothesis" and explain the concept.
- Line 67: There is no ‘lack of consistency’. I would say that there is ‘no consensus among the authors’.
- Lines 67-80: Please, authors do not use 1), 2), etc. Instead introduce the items in the following way: The first issue…, The second issue…
- Lines 70-72: It is not clear what the authors meant. Please, extend and rephrase.
- Lines 72-73: The sentences: ‘Only a few studies have analysed it from the perspective of the effects of environmental regulation. Therefore, five indicators were selected:’ are not well connected. Apart from that, I think that first the authors should present the three issues in a concise way. After that, they should focus on the indicators that they select and explain clear why they are selected (i.e. based on which criteria). Of course, they can connect they indicators to the second issue. At the moment everything is mixed and the text does not read well.
- Lines 82-84: This is an important piece of text. It should be treated as a separate paragraph and extended. The authors should clearly show how their choice permits to overcome the knowledge gap in the existing literature.
- Lines 84-95: This should become a new paragraph.
- Lines 96-99: Please, do not use ‘second part’, ‘third part’, etc. Simply use ‘Section 2’, ‘Section 3’.
- Line 102: Please, check the sentence. It looks like there are some redundant words.
- Lines 103-104: Here the authors introduce tow important concepts: ‘GML productivity index’, ‘non-radial slack based measure (SBM) directional distance’. This needs some explanations and references so that a general audience can understand it.
- Lines 140-145: ‘Selected’ is used many times.
- Line 145: Use ‘data’. ‘research data’ is redundant.
- Line 171-186: Please, use present and not past tense. Example: ‘The first step is’.
- Section 2 (Data and Methods) and Section 3 (Research design) should be restructured. I propose to have Section 2 (Methodology) and Section 3 (Data). Section 2 will include old sections: 2.1 Measurement of GTFP; 2.2 Calculation of the environmental regulation intensity; 3.1 Construction of the measurement model (this one will become 2.3). Then, Section 3 will have the following structured: 3.1 Sample and sources (this will be old section 3.2). Now, a new section will be included: 3.2. Facts. The new sub-section ‘3.2. Facts’ should present Figures 1 and 2, along with their explanations. In this way, the theory is presented separately from the empirical evidence.
- Line 222: Please, edit the heading using the same style as the other headings in the manuscript.
- General comment applicable to the entire manuscript: Please, write the manuscript using the present tense and not the past tense. At the moment both tenses are mixed in a not consistent way.
- Lines 404-406: ‘The government has imposed strict environmental regulations to force companies to reduce pollution emissions.’ This is an important statement that needs proper referencing and further explanation.
- Line 411-412: ‘Furthermore, the implementation of environmental regulations has promoted technological 411 progress in various enterprises’. This statement needs to be supported with examples or a reference.
- Lines 467-479: The text should be condensed.
- In section 6, the authors should clearly highlight which are the policy implications of their research. Which lessons could be extrapolated to other cities? The authors made a start with this in lines 480-485. However this needs further extension.
- Further editing of the text would be beneficial – sometimes the language is a bit redundant, there are some typos, etc.
Round 2
Reviewer 1 Report
The modification on "Research design" and "Empirical results" are good.
The narrative of section "2.3 Data source" still need to be revised.
Reviewer 4 Report
In general terms, the manuscript has notable improved. I would like to recommend publication after the authors correct some minor issues.
- Line 207, 226, 968: Please, check the spacing between words.
- Line 1002: Please, use 'policy recommendations' instead of 'policy enlightenments'.
- General comment: Please, authors go again through the entire manuscript. There are some minor typos that should be corrected (e.g. spacing issues in the lines mentioned above).